# OpenReview forum: "A transfer learning framework for weak to strong generalization"
_ICLR.cc/2025/Conference — ICLR 2025 Poster_

### Official Review · Reviewer_DyBd · 2024-10-31

**Soundness:** 2
**Presentation:** 3
**Contribution:** 2
**Rating:** 6
**Confidence:** 3

**Summary:**

The paper introduces a method to improve weak-to-strong generalization using labels from the strong model provided ICL examples from the weak model. They also provide a theoretical analysis of weak-to-strong generalization on a regression task.

**Strengths:**

The idea of using ICL examples to refine labels for weak to strong generalization is novel, and the paper provides theoretical analysis and motivation for the method. They also perform experiments on multiple common benchmarks and find improvements over naive fine-tuning on weak labels.

**Weaknesses:**

For baselines, there are other existing methods beyond naive fine-tuning such as using an auxiliary confidence loss or using intermediate models as presented in the weak to strong paper (Burns et al., 2023) that aim to address similar issues. It is unclear how this method compares to these existing methods given that they have the same goal. It would be helpful if comparisons to using the auxiliary confidence loss are added.

**Questions:**

1. How does this method compare to existing methods for weak to strong that aim to address limitations of naive fine-tuning?

---

> ### Author Response · Authors · 2024-11-16
> **Response to Reviewer DyBd**
>
> Thank you for the suggestions on empirical results to add. Below we include (our best efforts to recreate) the mentioned OAI baselines on the persona experiment.
>
> For bootstrapping we use the model chain weak -> gpt-3.5-turbo -> gpt-4o-mini. Unfortunately, it is not possible for us to directly implement the auxiliary loss method of OAI since we cannot access the model weights to train with a custom loss. As a proxy, we used a data doubling method, where each training question has an answer from the weak teacher and an answer produced by the strong model. We believe this is a reasonable way to emulate the auxiliary loss method, which adds a term that punishes the model for deviating from its own predictions. If the reviewer has a different idea in this vein they wish us to test, we are happy to. If, on the other hand, this is satisfactory, then we will add these results and discussion to the submission.
>
> Our findings are below,  "cont" denotes the accuracy while "style" measures the amount of concept transfer. Bootstrapping offers some safeguarding against accuracy degradation, but generally still allows for too much corruption of the strong model. The auxiliary loss improves accuracy but reduces the style transfer (this is expected in light of our theory, one can think of this method as strengthening $\eta$ in Prop 3.2). Overall, our methods offer a nice balance of (mostly) transferring the concept with minimal degradation.
>
> TTQA:
>
> |  | llama | gemma| mistral | Falcon |
> |----------|-------------|-----------|-------------|-----------|
> |Weak Training|cont: 3.54/ style: 8.51| cont: 4.55/ style: 8.4 | cont: 2.78/ style: 7.937 | cont:2.29/ style: 4.583  |
> | Aux_loss_prox| cont: 7.921/ Style: 2.377 | cont: 6.45/style: 5.65 | cont: 7.18/ style: 3.57 | cont: 6.141/style: 3.57   |
> | Bootstrap | cont: 3.79/ style: 8.482  | cont: 5.289/style:8.47 | cont: 3.12/style: 7.87 | cont: 2.883/style: 4.49  |
> | ICL (ours) | cont: 7.33 /style: 7.99 | cont: 7.069/style: 7.069| cont: 7.071/style: 7.846 |cont: 7.58/style: 5.703|
> |ICL + Sys Prompt (ours)| cont: 6.354/ style: 9.093 | cont:6.484/style: 9.036 | cont: 6.426/style: 9.149 | cont: 6.57/style: 9.07 |
>
> TAE:
>
> |  | llama | gemma| mistral | Falcon |
> |----------|-------------|-----------|-------------|-----------|
> |Weak Training| cont:5.6/ style: 8.72 |cont: 6.202/ style: 8.572| cont: 3.73/ style: 7.81 | cont: 3.266/ style: 4.512 |
> | Aux_loss_prox| cont: 8.577/ style: 2.42 | cont: 8.33/ style: 3.141 | cont: 8.26/ style: 2.74 | cont: 7.409/ style: 2.837   |
> | Bootstrap | cont: 6.14/style: 8.649  | cont: 7.028/style: 8.306 | cont: 4.59/style: 7.30 | cont: 3.191/style: 3.662  |
> | ICL (ours) | cont: 8.08/style: 6.707 | cont: 8.12/style: 6.258 | cont: 8.267/style: 5.74 | cont: 8.195 style: 4.014|
> |ICL+Sys Prompt (ours)|cont:7.808/ style: 8.63|cont: 7.499/style: 9.0|cont: 7.64/ style: 8.76| cont: 7.79/style: 8.26 |

---

> > ### Author Response · Authors · 2024-11-21
> >
> > With the discussion period ending in a few days we want to ensure that we have time to add any new empirical results that you believe would improve the paper. Are there any remaining concerns you have about the updated draft or new empirical results?

---

> > > ### Comment · Reviewer_DyBd · 2024-11-22
> > >
> > > Thank you for providing the empirical results and I will update my score to 6.

---

> > > > ### Author Response · Authors · 2024-11-22
> > > >
> > > > Thank you for the recognition and helpful suggestions!

---

### Official Review · Reviewer_Gdt9 · 2024-11-10

**Soundness:** 3
**Presentation:** 3
**Contribution:** 2
**Rating:** 6
**Confidence:** 3

**Summary:**

This paper mainly studies the problem of weak-to-strong generalization, that is, how to use the feedback from relatively weaker models to train and align more powerful language models without compromising their capabilities.

**Strengths:**

The paper proposes a theoretical framework that transforms the weak-to-strong generalization problem into a transfer learning problem, and proves that weak-to-strong generalization is feasible under this framework.

**Weaknesses:**

The framework assumes a convex hull relationship between the source model and target distribution, which may be too idealistic in practical applications. This convex hull assumption implies that the target distribution (the distribution that the strong model aims to achieve) must be covered by a combination of the source model's distributions. This is a strong theoretical assumption because in practice, stronger models may produce outputs that are completely beyond the capabilities of weaker models.

**Questions:**

1. In practical applications, how can we determine whether there truly exists a convex hull relationship between the source model and target distribution?
2. If we find that the source model cannot fully cover the target distribution, what are some feasible solutions?

---

> ### Author Response · Authors · 2024-11-16
> **Response to Reviewer Gdt9**
>
> Thank you for the helpful comments and questions. We discuss specific concerns in the following.
>
> *in practice, stronger models may produce outputs that are completely beyond the capabilities of weaker models.*
>
> We wish to clarify that our framework allows for this to occur. While the target model must be in the convex hull of the source model, the weak model **does not** have to satisfy this property. For example, the weak model may be a version of the (aligned) strong model that is missing some features or underfit in some way.
>
> *strength of convex hull relationship and how we can determine it in practice*
> We use the convex hull assumption to encode the intuition that the strong source model might possess some latent ability to accomplish the target task; it needs only to be directed to do so by the weak model. This assumption allows us to reason statistically about methodologies to try, but is not a pre-requisite for implementing our refinement methods.
>
> Practically, our suggestion is to try our refinement approach, and if it fails, it may suggest that the assumptions that lead us to this approach do not hold.
>
> *If we find that the source model cannot fully cover the target distribution, what are some feasible solutions?* In light of the above discussion, we believe the heart of this question is enquiring about settings where our methods can fail. We agree that weak to strong generalization is an on-going open problem, some alternative methods are suggested in the references below, but they are certainly not uniformly successful (and several are extensions of our simple ICL method or baselines we beat in our experiments).
> [1] Weak-to-Strong Generalization: Eliciting Strong Capabilities With Weak Supervision. 2023
> [2] Yang et al. Weak-to-Strong Reasoning. 2024

---

### Official Review · Reviewer_tWWA · 2024-11-11

**Soundness:** 3
**Presentation:** 3
**Contribution:** 3
**Rating:** 8
**Confidence:** 3

**Summary:**

This submission studies weak-to-strong generalization, where a weaker but aligned model is used to align a stronger but unaligned model. It is commonly assumed that weak-to-strong generalization is possible because the weak model supervision elicits knowledge that is already captured by the strong model but is possibly not expressed with desired frequency. In a similar spirit, the authors consider a theoretical setting, where both weak and strong models are assumed to have conditional probability density (of scalar output Y given a prompt X) that is of the following form:
$$
p(Y|X) = \sum_{k} \alpha_k \mathcal{N}(y; \beta_k^T \phi^*(X), \sigma^2),
$$
where $\phi^*(X)$ is some *observed* representation that is assumed to be the same for both models, while $\alpha_k$ and $\beta_k$ are *unobserved*, with $\beta_k$ representing concepts and $\alpha_{1:k}$ being a prior distribution over concepts. In this framework, the authors formulate weak-to-strong generalization as transferring weak teacher’s prior probabilities $\alpha_{1:k}$, while assuming one of the following conditions:
* Noisy weak model: $\beta_k$ are shared by the strong and weak models, but the labels of the weak model are noisy.
* Biased weak model: $\beta_k$ of the weak model are corrupted versions of target aligned strong model’s $\beta_k$.

One of the main contributions of this submission is demonstrating that naively fine-tuning the strong model with labels sampled from the weak model can lead to quality degradations (Theorem 3.2). Furthermore, with experiments persona learning, mathematical reasoning, and explanation technique learning, the authors show that this simple fine-tuning strategy leads to content quality drop, even though alignment (i.e., style transfer in case of personal/explanation tehncique learning) is often successful.

To improve weak-to-strong generalization, the authors propose a method that uses in-context learning capabilities of the strong model to refine the weak model’s labels (e.g., make them factually more correct) while adhering to the “style” presented to the strong model with a few in-context examples. These refined labels are then used to fine-tune the strong model. Within the proposed theoretical framework, the authors prove that under certain assumptions, this procedure can lead to a successful weak-to-strong generalization. Empirically, the proposed algorithm often manages to retain the strong model's quality, while transferring the style of the aligned weak model significantly, although not as well as the naive fine-tuning does.

**Strengths:**

* The paper is generally well-written and the presentation is clear. See my comments below for minor suggestions.
* The proposed theoretical framework captures some essential aspects of weak-to-strong generalization, while being amenable to analysis. The theoretical results proved in this framework are good contributions to the theory of weak-to-strong generalization.
* The authors explore a few techniques to improve upon fine-tuning with weak model’s labels. The proposed method and the alternative variants presented in appendices B and C are practical enough and can be helpful for designing better methods in the future.

**Weaknesses:**

**Strong assumptions that are sometimes not elaborated enough.**
* Most importantly, the authors make strong assumptions on the form of the weak model, unaligned strong model, and the target (aligned) strong model.
* It is assumed that the weak model is perfectly aligned (i.e., its $\alpha_k$ are perfect)?
* Why do betas have to be orthonormal at Line 104? Also, as I understand, unit norm is not assumed for target distributions. Similarly, why is orthogonality needed in the second case of Assumption 2.1.
* It would be helpful to provide some intuition on the two cases of Assumption 2.1.
* Line 138: How important is it to assume identity covariance for features $\phi^*(X)$?

**Insufficient exploration of simpler weak-to-strong generalization techniques than the proposed one summarized above.** Besides the main approach (label refinement with ICL and then fine-tuning on refined labels), the authors propose two simpler strategies in appendices B and C. Furthermore, another simple strategy is to not fine-tune the strong model, but just use its in-context learning ability (possibly with a suitable system prompt) to produce a prediction on a query given a few in-context demonstrations from the weak teacher. To better understand whether the complexity of the proposed approach is needed, it would be great to explore these simpler approaches more (e.g., better prompt engineering)  and compare them to the main proposed technique explicitly in a joint future or table.

**Questions:**

* Line 117: $X_q$ should be just $X$. Also, one of the $\sigma^2$ is extra.
* Lines 139-141: It would be helpful to state the excess loss explicitly as MSE between $\mathbb{E}_Q[Y|X]$ and $\hat{\beta}^T\phi^*(x)$.
* Line 142: “The subsequent output is an example of source and target priors over the concepts and a weakly supervised sample” – this sentence is unclear.
* Lines 200-204: It should be $\alpha_1$ instead of $\alpha$.
* Theorem 3.2. Would be helpful to comment why there is no $\eta$ on the right hand side.
* Lines 303-308: This part needs more elaboration. It would be helpful to add a derivation of the first line.
In figures 1, 2, and 4, do the 4 weak models on the “Strong model content score vs Weak model content score” and “Strong model style score vs Weak model style score” subplots align with each other. In other words, are the weak models on both plots the same as we scan left to right?

---

> ### Author Response · Authors · 2024-11-16
> **Response to reviewer twwA**
>
> Thank you for the helpful comments and concerns. Below we address specific issues.
>
> *Strong assumptions that are sometimes not elaborated enough.*
>
> * The assumption that the weak model has the correct prior is likely necessary in theory; without this, the learner has no access to either the correct model or the correct prior over the concepts. Philosophically, this is also reasonable as when applying weak-to-strong generalization to achieve its main objective in practice, i.e., allow a human to align a strong LLM to their interests, it is a natural assumption since the human's prior is the prior they wish to transfer onto the LLM (i.e, human is "perfectly aligned").
>
> * We wish to clarify that the assumption that betas are orthonormal should hold in the target too. In all cases, the orthonormality is not necessary for the results, but it substantially simplifies the terms in bounds, generally improving interpretability. What is important is some distinguishability between the $\{\beta_k\}$, orthonormality is a simple way to impose this.
>
> * Likewise, identity covariance of $\phi^*$ is not necessary. In the revised draft we have simplified things by making population assumptions on the features rather than fixing a design matrix.
>
> * The assumptions in case 2.1 were meant to represent simple cases where the weak model is some underfit (e.g. missing some features) version of the strong model (in practice, the weak model is indeed much smaller). These were not necessary, so we have dropped them from the results.
>
> *Insufficient exploration of simpler weak-to-strong generalization techniques than the proposed one summarized above*
>
> For a more thorough exploration of the first auxiliary method (from Appendix D), see the table in the response to reviewer DyBd. We have added more baselines and included a side-by-side comparison with the ICL method in the main text.
>
> We make a general note here that we don't believe one method is better than another; rather, these are multiple ways we thought of for implementing a refinement type procedure. The best choice is likely highly dependent on the task at hand.
>
> *Elaboration of Theorem 3.2.*
>
> We have simplified the lower bound in the new draft. Note that before $\eta$ did not appear because we minimized over $\eta$ for the final lower bound; in the new version, we have removed this final step and left the dependence on $\eta$ explicit.
>
> *Elaboration of lines 305-308*
> In order to clarify the presentation and provide a better connection to prior work on statistical models for LLM's we have made changes to the presentation of this area. Namely
> (i) We have laid out directly the assumption we need for Theorem 4.3 and in appendix A added two references for prior works ([1], [2]) where this assumption holds.
> (ii) To appendix B, we have added a calculation for the case where this assumption does not hold.
>
> [1] Xinyi Wang et al. Large Language Models Are Latent Variable Models: Explaining and Finding Good Demonstrations for In-Context Learning. 2023
>
> [2] Reese Pathek et al. Transformers can learn regression mixture models. 2024.
>
> *Clarification of figures*
>
> Yes, the weak models (llama 7b, gemma 2b, falcon 7b, mistral 7b) remain constant left to right. In a given row of plots the order goes gpt-3.5-content, gpt-3.5-style, gpt-4o-mini content, gpt-4o-mini style. Note here gpt is the strong model.
>
> In the case of the math experiments, no labels from falcon 7b are used.

---

> > ### Comment · Reviewer_tWWA · 2024-12-02
> >
> > Thank you for the detailed response. I am keeping my original score.

---

### Official Review · Reviewer_H358 · 2024-11-12

**Soundness:** 3
**Presentation:** 2
**Contribution:** 2
**Rating:** 6
**Confidence:** 2

**Summary:**

This work studies weak-to-strong generalization where labels from a weaker model are used to improve the capabilities of a stronger model. Unlike prior works which explain this with the superior extrapolation capabilities, or the ability of stronger models to self-correct incorrect labels, this work assumes a latent concept model as in Xie et al. (2021). Under this model, they re-state the weak-to-strong generalization problem as a transfer learning problem in which one wishes to transfer a prior over a latent concept from the weaker model to the stronger model. In the same model, they show that naive finetuning on weak labels leads to a predictor with poor expected risk, but their refined finetuning approach based on Yang et. al. (2024) runs an implicit Bayesian inference procedure that is able to illicit latent knowledge from the stronger model.

**Strengths:**

- To study weak-to-strong generalization, the paper proposes a transfer learning problem. Here, the model for Y|X induced by a strong unaligned LLM, and the model for Y|X induced by an aligned but weaker LLM (that produces labels) share a latent structure that allows for a cleaner analysis of the predictor returned by the alignment process.
- The result in Theorem 3.2 clearly shows that naively finetuning on labels from a biased weaker model can lead to performance no better than that of the weaker model.
- A large chunk of the analysis relies on the following assumption: the aligned Q(Y|X) and unaligned P(Y|X) models sharing the same orthonormal basis, and this ensures that the optimal predictor $E_Q[Y|X]$ is contained in the convex hull of the source distribution. While this assumption is limiting and unclear if true in practice, it presents a clean explanation of how recovering the latent structure from Y|X can fix the labels from the weaker LLM, and thus provably providing a way to generalize from weaker models, as illustrated in an idealized setting in Section 3.2.
- Using ICL to correct weaker labels is a simple and empirically effective way to correct weak labels, as demonstrated by their experiments on math, tinyAlpacaEval, and tinyTruthfulQA.

**Weaknesses:**

- The connection between the theoretical model and practical analysis is weak.
  - E.g., it is unclear if ICL is actually performing implicit Bayesian inference under the assumed model. In fact, this is assumed almost
     directly from the claims in Xie et. al. (2021).
  - The assumptions (Assm 2.1) made to model the bias in weakly aligned models is not well motivated. They may be amenable for
     theoretical convenience, but there is no reason to believe that the biases behave as nicely as assumed by the model.
  - The analysis assumes that the source (unaligned LLM) and target (aligned and correct LLM) distributions share the same convex hull
     realized by the fixed set of $\beta$s. It is unclear if this is true in practice, since this is equivalent to saying that the weaker LLM learns
     almost all concepts equally well during pre-training, and any errors in the weaker LLM are covered by Assumption 2.1.
- The empirical results are missing some key baselines. If we ignore the labels from the weak model, and directly use the stronger model to label data, e.g. with CoT prompting for MATH, would that curate a good dataset too? Currently, it is unclear if the weaker model is actually enabling implicit Bayesian inference.
- As the performance of the weak model improves (still subpar compared to stronger model), naive finetuning outperforms their ICL version of finetuning in multiple tasks. This makes their approach effective only when the labels are highly biased. In this case, I wonder if we can throw away the weaker model, and simply self-train on the stronger one.

**Questions:**

- It is unclear how $\kappa$ or the strength of the aligned LLM changes the error lower bound in the second part of Theorem 3.2. Can the authors please expand on the discussion here?
- What does “labels from one source” mean in L216?
- How does the separability assumption on the latent clusters sidestep the lower presented in Theorem 3.2 (L217)?
- How tight is the result in Theorem 4.2? It seems that $n_{ICL} = o(1/\rho)$ for the result to be non-trivial. Is ICL or some form of Bayesian inference on latent concepts, the best one can do in the worst case?

---

> ### Author Response · Authors · 2024-11-16
> **Response to reviewer Reviewer H358**
>
> Thank you for the helpful comments, we address your concerns below.
>
> *The connection between the theoretical model and practical analysis is weak.*
>
> We agree that the validity of our transfer learning framework is dependent on prior results demonstrating that ICL can be modeled as implicit Bayesian inference. In order to strengthen the connection between theory and practice, we have done the following:
>
> (i) To appendix A We have added discussion on two more references that consider latent bayesian/ mixture models of LLM's. The first [1] proposes a simplified version of Xie et al's work. The second [2] shows that LLM's can directly learn the mixture of regression setting we work in.
>
> The theory in our work can be seen as specifying an $f$ in [1] and studying ICL inference when the examples are ``weak". If the latent concept model is less convincing, an alternative interpretation of the theory is an analysis of the MSE of labels produced by the architecture in [2] (with the complication that examples are produced by a weaker model).
>
> (ii) To provide a more direct connection to Xie et al, we have now included an analysis where the source model is a mixture of Hidden Markov Models. This is an extension to the main result of Xie et al where ICL examples come from a weaker model. This setting is a bit more complex but is slightly closer to a real LLM (in particular delimiter tokens can be treated formally).
>
> [1] Xinyi Wang et al. Large Language Models Are Latent Variable Models: Explaining and Finding Good Demonstrations for In-Context Learning. 2023
>
> [2] Reese Pathek et al. Transformers can learn regression mixture models. 2024.
>
> *Improving assumptions made to bias the weak model*
>
> These are not strictly necessary for the analysis, and we have dropped them from the results in the paper.
>
> *empirical results are missing some key baselines.*
>
> (i) "If we ignore the labels from the weak model, and directly use the stronger model to label data...would that curate a good data set too?"
>
> The motivation of the weak-to-strong generalization setting is some form of (weak) human providing examples to teach the model. So we have chosen to focus on methods which utilize the human/weak model data in some manner. The necessity of this data is clear in the style experiments where throwing away all weak data will clearly lead to poor performance, as there will be nothing left influencing the strong model towards the desired style.
>
> Beyond this, completely ignoring the weak model and generating synthetic data from the strong model to improve itself is out of scope of the weak-to-strong generalization problem setting. It has different goals (i.e. improving the model's overall capabilities through its own synthetic data, as opposed to aligning the model to the specific interests of a weaker human teacher) and many prior works have already studied this problem, e.g.
>
> Wang et al. Self-Instruct: Aligning Language Models with Self-Generated Instructions, 2023.
> NVIDIA, Nemotron-4 340B Technical Report, 2024
>
> That said, we do agree that the use of some plain self-training data (along with weak data) is an interesting approach. In the response to reviewer DyBd, we have included a comparison to a method which mixes in such data.
>
> (ii) "As the performance of the weak model improves (still subpar compared to the stronger model), naive finetuning outperforms their ICL version of finetuning in multiple tasks"
>
> While it is true that for some weak models, naive fine-tuning can improve the style transfer, we disagree that this should be interpreted as naive fine-tuning outperforming our ICL method. As an example, consider the explanation technique experiment. While naive fine-tuning leads to simpler explanations, the explanations themselves become incorrect as the strong model capability is reduced. In general, we do not interpret this as a more successful outcome. Finally, we also note that extensions to our method discussed in the appendix eliminate this "style gap".
>
> *Other Questions*
>
> * We have simplified the lower bound in the new draft so that the role of the bias of both the source and weak models is clear.
>
> * Labels from one source refer to labels from only one weak teacher. We have clarified this in the draft.
>
> * Statistically, the lower bound is avoidable because inferring the target discrete concept from the weak data is easier than trying to learn the target model. The separability makes this inference possible.
>
> * We agree that the tightness of theorem 4.2 is an interesting future question. To address this properly would require establishing the minimax rate of estimating $f_Q$ in our set up. This is of interest to us but is non-trivial, and we hope to consider it in future work. It is quite possible that our method is not optimal in the minimax sense. That said, the statistically optimal method likely does not have any analog that can be implemented for a real LLM, which is one advantage of our refinement-based estimates.

---

> > ### Author Response · Authors · 2024-11-21
> >
> > With the discussion period ending in a few days we want to ensure that we have time to add any new empirical or theoretical results that you believe would improve the paper. Are there any remaining concerns you have about the updated draft or new empirical results?

---

> > > ### Comment · Reviewer_H358 · 2024-11-30
> > >
> > > I think the authors for their detailed responses, especially on adding more extensions like mixture of HMMs. After reading through the revisions, and comments from other reviewers, I feel that the paper does use the setup and algorithm in Xie et. al., but does go beyond the prior works and extends the implicit Bayesian inference argument to understand weak-to-strong generalization. While this is a neat connection, it is unclear if the theoretical contribution in and of itself is non-trivial enough. At the same time, I feel the authors supplement it with some valid empirical arguments. So I am slightly ambivalent but will lean towards acceptance. Thus, I am happy to raise my score to 6, but with a lower confidence of 2.

---

> > > > ### Author Response · Authors · 2024-12-01
> > > > **Thank you**
> > > >
> > > > Thank you for taking the time to read our added results and update your score!

---

### Author Response · Authors · 2024-11-16
**Important changes in new draft**

Thank you to all reviewers! Below is a list of important changes we have made for the new draft.
* In the transfer learning setup (section 2), we have assumed that features are random (rather than fixing a design matrix).
* We have removed assumption 2.1, $\beta_k^w$ can now take a general form.
* We have simplified the lower bound in Theorem 3.2 (note this is replaced by proposition 3.2 in the new draft).
* We have made changes to the presentation of our theoretical analysis of ICL refinement (section 4.1.1). In particular, we have divided the analysis into two cases: the first is where the model treats the ICL examples as iid, the second is where it does not. Appendix A contains discussion on when the first case holds. Appendix B contains a treatment of the second case.

---

### Meta-Review · Area_Chair_Uxwa · 2024-12-23

**Metareview:**

(a) The paper claims to prove the possibility of weak-to-strong generalization by eliciting latent knowledge from pre-trained LLMs. It casts this problem as a transfer learning problem, aiming to transfer a latent concept from a weak model to a strong pre-trained model. Theoretically, it shows that naive fine-tuning on weak labels can lead to poor expected risk, while a refinement-based approach can overcome this. Empirically, the paper demonstrates the practical applicability of the refinement approach in multiple LLM alignment tasks such as persona learning, mathematical reasoning, and explanation technique learning.

(b) Strengths:
  - The paper proposes a novel theoretical framework that provides a new perspective on the weak-to-strong generalization problem, which is a significant contribution to the field.
  - The empirical results on various tasks support the proposed method and show its effectiveness in improving weak-to-strong generalization compared to naive fine-tuning.
  - The idea of using in-context learning (ICL) to refine labels is innovative and offers a practical solution to address the challenges in this area.

(c) Weaknesses:
  - The theoretical assumptions, such as the convex hull relationship between models, are relatively strong and might not hold in all practical situations.

(d) Reasons for Acceptance:
  - The paper addresses an important and timely problem in the field of LLM research, and the proposed solution has the potential to impact future research and practice.
  - The authors have made significant efforts during the rebuttal period to address the reviewers' concerns. They have provided additional theoretical explanations, dropped unnecessary assumptions, and conducted more detailed empirical comparisons.
  - Despite the weaknesses, the overall quality of the paper in terms of novelty, methodology, and empirical validation is sufficient to merit acceptance. The work offers new insights and a practical approach that can inspire further research in this area.

**Additional Comments On Reviewer Discussion:**

(a) Reviewer Points and Author Responses:
  - Connection between Theory and Practice: Reviewers questioned the link between the theoretical model and the practical use of ICL. The authors added discussions in the appendix, referring to other works that support the idea of LLMs as latent variable models and the connection to ICL inference. They also provided an alternative analysis with a more complex source model (mixture of Hidden Markov Models) to strengthen the connection.
  - Model Assumptions: Concerns were raised about the assumptions on the weak model, such as the correct prior and orthonormality. The authors clarified that the correct prior assumption is theoretically necessary and that orthonormality simplifies the analysis but is not essential. They also removed some assumptions that were not crucial for the results.
  - Simpler Techniques Exploration: Reviewers noted the lack of exploration of simpler methods. The authors added more details on an auxiliary method in the appendix and compared it with the ICL method in the main text. They emphasized that different methods may be suitable for different tasks.
  - Empirical Results: Reviewers asked for more baselines and comparisons. The authors added empirical results comparing their method with a proxy for the auxiliary confidence loss method and other baselines in the persona experiment.

(b) Weighing in the Final Decision: The authors' responses to the reviewers' concerns were comprehensive and effective. They addressed each point with additional explanations, theoretical justifications, and empirical evidence. The improvements made during the rebuttal period enhanced the paper's overall quality and convinced that the paper's contributions outweigh its weaknesses. The novelty of the proposed framework and the practical applicability shown in the experiments were also important factors in the decision to accept.

---

### Decision · Program_Chairs · 2025-01-22

Accept (Poster)